# Large-Scale Particle Image Velocimetry to Measure Streamflow from Videos Recorded from Unmanned Aerial Vehicle and Fixed Imaging System

**Wen-Cheng Liu** [1,*], **Chien-Hsing Lu** [1] **and Wei-Che Huang** [2]

1   Department of Civil and Disaster Prevention Engineering, National United University, Miaoli 36063, Taiwan;
    M0615006@gm.nuu.edu.tw
2   College of Engineering and Science, National United University, Miaoli 36063, Taiwan;
    d0412002@gm.nuu.edu.tw
*   Correspondence: wcliu@nuu.edu.tw; Tel.: +886-37-382357

**Abstract:** The accuracy of river velocity measurements plays an important role in the effective management of water resources. Various methods have been developed to measure river velocity. Currently, image-based techniques provide a promising approach to avoid physical contact with targeted water bodies by researchers. In this study, measured surface velocities collected under low flow and high flow conditions in the Houlong River, Taiwan, using large-scale particle image velocimetry (LSPIV) captured by an unmanned aerial vehicle (UAV) and a terrestrial fixed station were analyzed and compared. Under low flow conditions, the mean absolute errors of the measured surface velocities using LSPIV from a UAV with shooting heights of 9, 12, and 15 m fell within $0.055 \pm 0.015$ m/s, which was lower than that obtained using LSPIV on video recorded from a terrestrial fixed station (i.e., 0.34 m/s). The mean absolute errors obtained using LSPIV derived from UAV aerial photography at a flight height of 12 m without seeding particles and with different seeding particle densities were slightly different, and fell within the range of $0.095 \pm 0.025$ m/s. Under high flow conditions, the mean absolute errors associated with using LSPIV derived from terrestrial fixed photography and LSPIV derived from a UAV with flight heights of 32, 62, and 112 m were 0.46 m/s and 0.49 m/s, 0.27 m, and 0.97 m/s, respectively. A UAV flight height of 62 m yielded the best measured surface velocity result. Moreover, we also demonstrated that the optimal appropriate interrogation area and image acquisition time interval using LSPIV with a UAV were $16 \times 16$ pixels and 1/8 s, respectively. These two parameters should be carefully adopted to accurately measure the surface velocity of rivers.

**Keywords:** unmanned aerial vehicle (UAV); LSPIV; flight height; seeding artificial particle; interrogation area; image acquisition time interval

## 1. Introduction

Surface flow is an important variable in hydrological river observations and can be used to further estimate the corresponding discharge. Generally, velocity observations are limited to fixed stations or fixed areas, and researchers are exposed to river measurement risks, which may endanger human life [1].

To reduce the risk of field observations, several studies have utilized photogrammetric techniques to measure the surface velocity of rivers. This related state-of-the-art technology has been continuously developed over many years [2–7], and with the advancement of technology, there are more image-based approaches that can be employed for surface velocity measurement. Through image monitors, television screens [8], or even videos from the internet [9], photogrammetry can be used to measure the surface velocity of rivers from pictures. Among the various video shooting vehicles available, unmanned aerial vehicles (UAVs), which possess highly maneuverable characteristics, can overcome terrain

constraints, have orthographic capabilities, and provide high economic benefits (i.e., low cost), are being favored by an increasing number of researchers [10–20]. Because UAVs can overcome terrain obstacles and perform aerial photography operations, UAVs have been widely utilized in agricultural and ecological surveys [21–25], environmental monitoring projects [26–29], disaster monitoring efforts [30–33], and other tasks [34].

Among the available optical techniques, large-scale particle image velocimetry (LSPIV) obtains remote surface flow measurements based on digital images obtained from a fixed location [2,35]. LSPIV can be applied to yield the surface velocity of the field, and the shooting range is larger than that of traditional PIV. However, issues that are encountered in the field that would not be faced in a laboratory setting need to be addressed; for example, the camera shooting angle is not guaranteed to be perpendicular to the water surface. It is necessary to find multiple ground control points (GCPs) for image calibration and orthorectification. LSPIV also requires the optical correction and elimination of camera lens distortion [36]. Compared to the use of portable surface velocity radars (SVR) [37], optical methods used for flow analysis [38,39], as well as other noncontact measurement methods used to determine surface velocity, LSPIV field measurement still has optical obstacles, even considering its flexible application to different environments without the restriction of instruments, such as the presence of reflections, shadows, and low light sources, that need to be overcome [7,40]. However, several studies conducted via field experimentation have verified that the surface velocity data measured by LSPIV display similar results to those measured by acoustic Doppler current profilers (ADCPs) [41,42]. This indicates that LSPIV methods possess sufficient accuracy and can be trusted to measure the surface velocity of rivers.

LSPIV also provides a flexible and cost-effective flow measurement method, which can effectively circumvent the deficiencies of river flow monitoring networks, and can be used to monitor sudden flash floods [43]. In addition to the river monitoring images provided by fixed specially erected monitoring stations [44,45], LSPIV can also be applied to existing surveillance video images taken in cities [8,46]. This supports the use of LSPIV in surface flow-based urban flooding and flood spread monitoring applications [47]. Moreover, Detert [48] investigated how to avoid and correct errors when applying LSPIV. He demonstrated that the intrinsic and extrinsic calibration parameters, acquisition frame rates, image and velocimetry filters, and bias reduction methods applied at near-zero velocities greatly improved the accuracy of the output, yielding much better agreement with the results of field measurements. Various researchers [49,50] have suggested employing Monte Carlo simulations to both estimate the uncertainty of the obtained LSPIV estimations and eliminate the subjectivity on the selection of the LSPIV algorithm parameters (e.g., interrogation area size, size of the contrast-limited adaptive histogram equalization, etc.).

In a recent study by Tauro et al. [10] and Detert and Weitbrecht [51], a UAV combined with LSPIV was successfully employed to measure the surface velocity of a river. The compact aerial sensing platform yielded a ground-facing orthogonal camera, the use of which did not require image orthorectification. Subsequently, Bolognesi et al. [12], Koutalakis et al. [16], Dal Sasso et al. [52], Pearce et al. [53], and Strelnikova et al. [17] integrated UAVs and LSPIV to measure the surface flow velocity of rivers.

According to previous studies from the literature, the combination of UAVs and LSPIV instruments, which are more mobile and can perform vertical aerial photography, will become an emerging technology for monitoring river flow in the future. Some studies using UAVs have focused on different topics, including flight height [54], seeding particle placement [17,52,55,56], interrogation area (IA) [17,54], sampling frequency [54], image stabilization [10,54,57,58], and acquisition time [54,57]. For example, Lewis et al. [54] revealed that long intervals between images affected the measured surface velocities derived from PIV. When the time between image frames increased, the displacement of particle patterns was more difficult to catch. They also found that the size of IA was lower than $64 \times 64$ pixels, resulting in decreasing the accuracy of measurement.

However, few relevant studies have been completed in Taiwan. The objective of this study is to measure the surface velocity of the Houlong River, Miaoli, in central Taiwan using UAV and LSPIV instruments with an intrusive flow meter and float method to verify the UAV- and LSPIV-based measurement results. This study also emphasizes using images obtained via terrestrial fixed photography to measure surface velocity with LSPIV and compares the LSPIV measurement results with UAV orthographic images and oblique ground images. In addition to verifying the accuracy of UAVs combined with LSPIV in measuring river surface velocity, this study further explores the influence of different experimental conditions on the measurement results, including the selected UAV shooting height, artificial seeding tracer method, interrogation area (IA), and acquisition time.

## 2. Materials and Methods

### 2.1. Description of the Study Area

The Houlong River originates from Jialishan Mountain, which is located in Miaoli County. The river flows from east to west into the Taiwan Strait in Houlong Township (Figure 1a). The main stream of the Houlong River is 58.3 km long and has a drainage area of approximately 536.59 km$^2$. It is one of the main rivers in western Taiwan. The annual average rainfall in the Houlong River Basin is 1998 mm, the monthly average temperature is approximately 15–30 °C, and the monthly average evaporation amount is 84.6 mm. The rainy season lasts from March to September, while the dry season lasts from October to February. The Houlong River presents a terrain that is high in the east and low in the west. The topography of the upper river, accounting for approximately 87% of the total river length, is mountainous. Therefore, the river slope exhibits very steep and rapid flow. The average slope of the middle and lower reaches of the river is approximately 1/260, while the average slope of the upstream reach is approximately 1/160, so the population is concentrated along the lower reaches of the river.

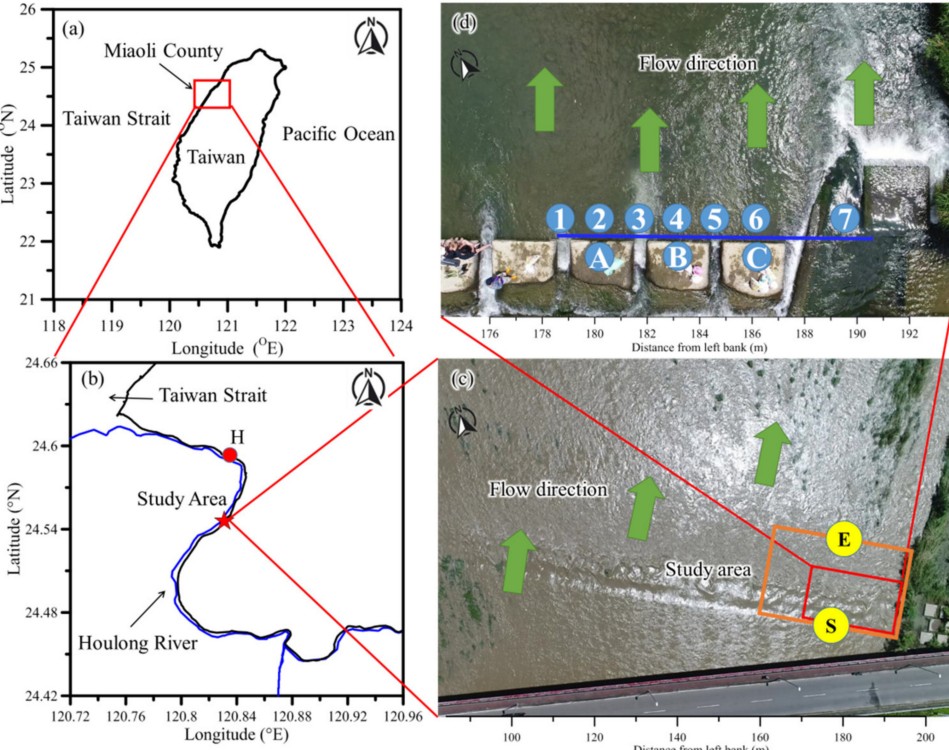

**Figure 1.** Map showing (**a**) the location of Miaoli County in Taiwan, (**b**) the study area located along the Houlong River, where the symbol H represents discharge station, and (**c**) the groundsill works flooded during high flow period. The symbols S and E denote the start point and end point using the float method, respectively, and (**d**) the groundsill works exposed during the low flow period. The measurement locations from number 1 to number 7 using a flow meter and the artificial tracers (i.e., wood shavings) released at locations A, B, and C are illustrated in the figure.

In this study, the Houlong River, which flows through the Xindong bridge in Miaoli City, was selected as the measurement station (Figure 1b). Miaoli is a densely populated area in the Houlong River Basin. Although the Houlong River flows through this area, it forms a shoal, and the riverbed is exposed to gravel and vegetation during the dry season. However, during heavy rainfall and typhoon events, the Houlong River experiences high flow conditions for a short period of time; therefore, riverside parks along both banks are occasionally flooded. Figure 1c,d display the study site where the groundsill works are flooded and exposed during the high flow and low flow periods, respectively. The river width of the study site is approximately 140 m. UAVs were utilized to take images of the river's surface, and the surface velocity was calculated by the LSPIV algorithm to verify this method of measuring the river surface velocity.

In this study, three field campaigns were conducted on 31 May, 15 June, and 10 December in 2019 to measure the river surface velocity. No rainfall occurred on 31 May or 10 December, while the average rainfall on the day before 15 June was 97 mm. The discharge rates on 31 May, 15 June, and 12 December were 15.9 cm$^3$/s, 69.0 m$^3$/s, and 3.0 m$^3$/s, respectively, at station H. Therefore, the flow conditions on 31 May and 10 December represent the low flow period, while those present on 15 June represent the high flow period. The wind speeds on 31 May, 15 June, and 10 December were 1.9 m/s, 2.2 m/s, and 1.2 m/s, respectively.

### 2.2. Terrestrial Fixed Imaging System

The fixed terrestrial imaging system is erected on the left rear of the study area on the bridge approximately 12 m above the surface water, as shown in Figure 2. The photography system adopts the color industrial camera ICDA-acA1600-20gc manufactured by Basler. The camera lens was equipped with a low distortion lens (FV1520), with a focal length of 15 mm, a maximum distortion of −0.09%, a frame rate of 20 fps (frames per second), and a captured image size of 1624 × 1234 pixels. The camera was placed in a protective case and then installed on a U-shaped frame. The U-shaped frame can be rotated horizontally and vertically to adjust the posture of the camera to shoot the desired field of view (FOV). The U-shaped frame was equipped with a level bubble for easy leveling. The captured image was sent to a computer via the transmission line, and the image grayscale was orthorectified using the RIVeR program. Then, PIVlab was utilized to calculate the river surface velocity (see Section 2.4).

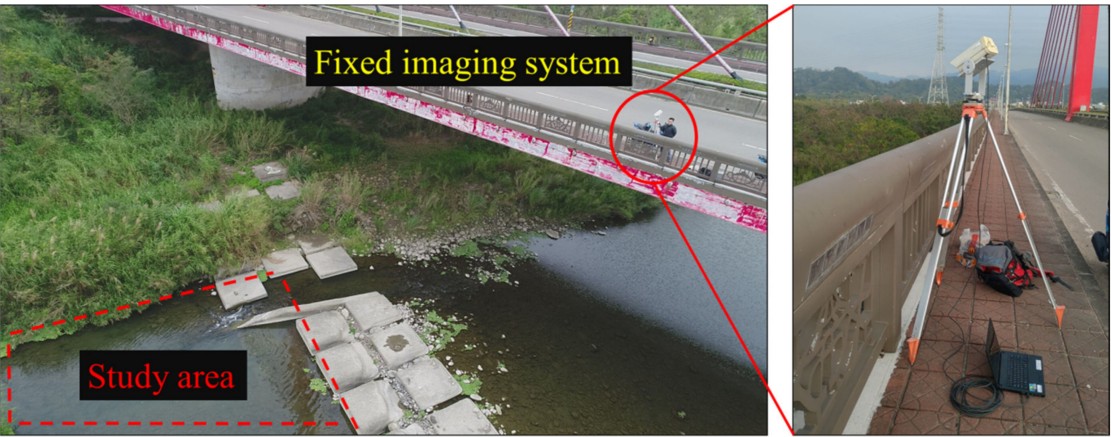

**Figure 2.** A terrestrial fixed imaging system was placed on a bridge to take images.

To make the shooting range of the camera cover the FOV, the shooting direction of the terrestrial camera and the direction of the river flow display a horizontal angle. The terrestrial fixed station also employs LSPIV to analyze images. Therefore, the surface velocities measured by UAV-based LSPIV and terrestrial fixed stations are also compared.

### 2.3. Unmanned Aerial Vehicle (UAV)

The model of the unmanned aerial vehicle used in this study is PHANTOM 4 PRO, which consists of a drone body, a remote control, a gimbal camera, and the DJI GO 4 program. The aircraft weighs 1388 g and has an FOV of 84 degrees and a maximum flying altitude of 500 m. However, the lighter airframe can only fly in environments with wind speeds below 10 m/s, and the hover range is 0–10 m. Images with the same shooting height should be taken within a short time frame to avoid external factors that could interfere with shooting.

To shoot images at the same level employing a UAV and terrestrial fixed photography, with a similar image quality and frame rate, a UAV photography image size of 1920 × 1080 pixels was selected, and the selected frame rate was the lowest available at 24 fps, making the terrestrial fixed photography image size 1624 × 1234 pixels; a frame rate of 20 fps would produce similar results.

When the UAV's flight altitude is higher, the ground distance represented by 1 pixel in the image increases. This may cause the calculated surface velocity to be somewhat larger and somewhat smaller than the real velocity. However, most of these errors can eliminate each other when calculating the average surface velocity.

### 2.4. Software PIVlab

To map the flow field of particles in a fluid, Thielicke and Stamhuis [59] utilized laser shooting, and a fixed camera was adopted to capture the movement of fluid particles. These images were input into PIVlab, written by Thielicke and Stamhuis in MATLAB. PIVlab can determine the highest similarity of particle groups between the previous image and the latter image based on the principle of image matching, and calculate the co-ordinates of the fluid particles. The calculated result is the most likely vector of the particle, which is divided by the time interval to obtain the speed [60,61].

PIVlab has been widely used by many researchers in image surveying and mapping. The software used for velocity analysis consists of three main steps: image preprocessing, image evaluation, and postprocessing. The image preprocessing techniques implemented in PIVlab include histogram equalization, intensity highpass filtering, and intensity capping. In this study, the fast Fourier transform (FFT) with multipass analysis is adopted for image evaluation. Postprocessing includes data validation, interpolation, smoothing, and exploration. The postprocessing is employed to display the measurement results of surface velocity. A detailed description can be found in Thielicke and Stamhuis [59].

In image preprocessing, the main step is to enhance the characteristics of the tracer in the images. The original image and the enhanced image are shown in Figure 3. There are more ripples on the right side of Figure 3b so, after preprocessing the image, the captured river surface ripples appear to be white, and the area on the right side of the image, where the flow velocity is faster, has more ripples. On the left side of the image, the flow velocity is slow, and the ripples on the river surface are not obvious, so the area on the left is black.

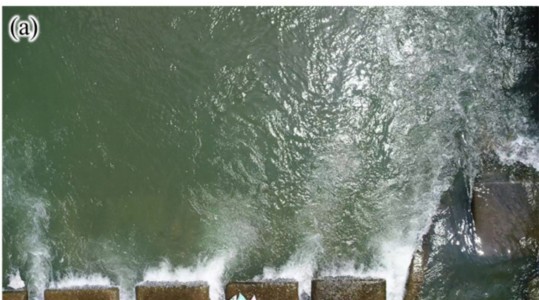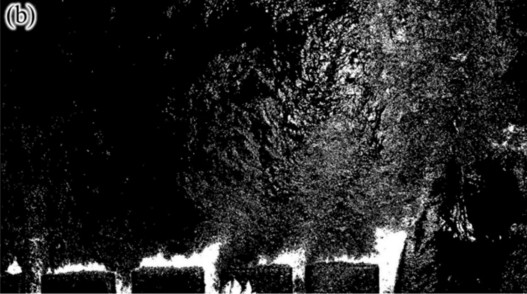

**Figure 3.** UAV shooting image on 31 May 2019; (**a**) original image and (**b**) image after preprocessing.

Image postprocessing mainly sets the *X*-axis and *Y*-axis directions of the velocity field. After velocity field analysis is completed in PIVlab, the velocity at each point is output for comparison with benchmark measurements.

In this study, the benchmark measurement of surface velocity is measured along the groundsill works (see Figure 1d). Based on the velocity analysis result obtained with PIVlab, a straight line is drawn along the groundsill works (see blue line in Figure 1d), and the velocity at each point along the straight line is output and compared with the measured velocities at the corresponding locations.

For each measurement, 8 shooting images are used for analysis. The original shooting frame rate is 24 fps. To yield better measurement results in the low flow region, the image acquisition frequency for analysis is 8 fps. After using PIVlab to analyze 8 images, the best analysis result is selected according to the relevance of each analysis result.

When UAVs are flying, they are influenced by wind movement and can shake as a result, and the three-axis gimbal of UAVs can eliminate part of the influence of wind speed [58]. For image shaking caused by wind speed, the additional program RIVeR can be used to correct the image shaking problem. Before running PIVlab, the RIVeR program was employed to convert the video shot by the UAV into a consecutive image and correct the image shaking problem. This means that image stabilization was performed.

### 2.5. Measurement Using Flow Meter and Float Method

In addition to the abovementioned UAV methods combined with the LSPIV method to measure surface velocity, this study also uses traditional river surface velocity measurement methods to verify the measurement results of the LSPIV method.

The employed propeller-type numerical flow meter is manufactured by the Global Water Company. The model type is FP111. Its shape is rod-like, with a total length of 1.22 m and a maximum extension of 2 m. Its measuring range is 0.1 to 6.1 m/s, and the measurement accuracy is $\pm0.1$ m/s. Because this flow velocity measurement method requires direct penetration into the water, it is classified as an invasive measurement instrument or a contact measurement instrument. The handheld flow meter takes measurements in 1-min intervals, and the instrument can automatically take the average of the data measured within one minute. The measurement locations obtained using a flow meter to measure the velocity are illustrated in Figure 1d.

When a higher water level occurs under rainfall conditions in the study area, the flow overflows the groundsill works. Considering the safety of the measuring personnel, the float method, which is a standard method, replaces flow meter measurements. The floats adopted in the field experiment are ping-pong balls, because the color of the ping-pong balls is in sharp contrast with the color of the water body. When the float is thrown into the river, it flows past the start point, and a member of the field team raises their right hand and records the time. When the person watching the float flowing through the end point raises his or her right hand, the time is recorded. The distance between the start point and the end point is 10 m (see Figure 1c). The distance between two sections is divided by the time difference between the floats passing through the two sections to obtain the average velocity. Three repetitions were measured for each float method.

### 2.6. Statistical Error

To quantify the error between the UAV method employing LSPIV/fixed stations with LSPIV and traditional methods of measuring surface velocity, the mean absolute error (MAE) and the root mean square error (RMSE) were adopted to compare the error results. The statistical errors can be expressed as:

$$MAE = \frac{1}{N}\sum_{i=1}^{N}|V_i^m - V_i^o| \qquad (1)$$

$$RMSE = \sqrt{\frac{1}{N}\sum_{i=1}^{N}[V_i^m - V_i^0]^2} \qquad (2)$$

where $N$ indicates the total number of measured values, $V_i^m$ denotes the measurement results obtained using the flow meter or float method, and $V_i^o$ expresses the measured values obtained using a UAV with LSPIV or a fixed station with LSPIV.

## 3. Measurement Results

Two field campaigns conducted on 31 May and 15 June in 2019 are described in this section. In each experiment, the surface velocity was measured by LSPIV with UAVs and terrestrial fixed stations and compared with the measurement results using the flow meter/float method. The purpose of experimental planning is mainly divided into three items:

1.  The measurement results using the LSPIV method with UAV aerial photography, the LSPIV method with a terrestrial fixed station, and the flow meter/float method were analyzed and compared. This means that the existing measurement methods used to measure the river surface velocity are utilized to validate the results using LSPIV derived by UAVs.
2.  In each experiment, a UAV was utilized to capture the river water surface at three different flight heights, and the measurement results were compared.
3.  The artificial particles were seeded on the water surface of the river, and the UAV was controlled to the height of terrestrial fixed photography, i.e., the height of the take-off location was 12 m, to take images under different artificial seeding particle densities and compare the measured results of river surface velocity.

In these two field campaigns, the IA window size and image acquisition time were set to 16 × 16 pixels and 1/8 s, respectively. The duration of image shooting was 1 s, and 8 frames taken within 1 s of image shooting were utilized for analysis.

### 3.1. Measurement Results on 31 May 2019 under Low Flow Conditions

The first experiment was conducted on 31 May 2019, under low flow conditions. The content of this experiment includes aerial photographs of UAVs at take-off heights of 9 m, 12 m, and 15 m, terrestrial fixed photographs, and flow meter measurements. In addition, different numbers of artificial particles (i.e., wood shavings) were soaked in water before introducing them into the flow at the same time to change the density of artificial particles flowing on the water surface using 1.5 kg per minute, 3.0 kg per minute, and 4.5 kg per minute. Figure 4 shows the shape and dimension of wood shavings used in the experiments. The area of wood shavings ranges from approximately 0.25 to 6 cm$^2$.

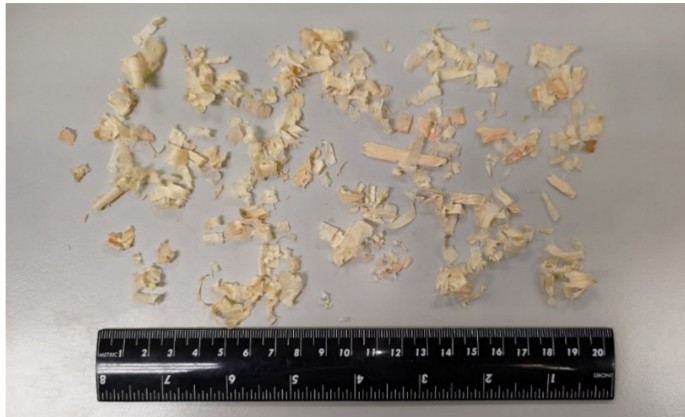

**Figure 4.** The shape and dimension of wood shavings regarded as artificial particles.

The size of the FOV at the physical scale, image resolution in cm/pixel, and IA window size of the PIV at the physical scale for different flight heights are illustrated in Table 1. The

measurement of the length scale is based on the square concrete block of the groundsill work. The length of the square concrete block is 2 m. Therefore, the two corners of the square concrete block are regarded as reference points for the scale of image pixel and length conversion.

**Table 1.** The ground sampling distance in each of the acquired image sequences.

| UAV flight height (m) | 9 | 12 | 15 | 32 | 64 | 112 |
|---|---|---|---|---|---|---|
| Resolution (cm/pixel) | 0.7 | 0.9 | 1.2 | 2.5 | 5 | 9 |
| IA coverage (cm × cm) | 10.2 × 10.2 | 14.4 × 14.4 | 19.2 × 19.2 | 40 × 40 | 80 × 80 | 144 × 144 |
| Image coverage (m × m) | 13.44 × 7.56 | 17.28 × 9.72 | 23.04 × 12.96 | 48 × 27 | 96 × 54 | 172.8 × 97.2 |

Figure 5 depicts the flow field generated by PIVlab from UAV flight heights of 9 m, 12 m, and 15 m. High velocity flow occurred between the groundsill works. In the figure, the distance from the left bank is taken as the *x*-axis, and the y direction is parallel to the river flow and represented as V (m/s). The surface flows of each checkpoint measured using a flow meter are presented on the *x*-axis and compared with the measured results using LSPIV taken at different UAV shooting heights, and the terrestrial fixed photography results are shown in Figure 6. In this figure, UAV_9 m, UAV_12 m, and UAV_15 m denote the LSIPV records taken from UAV flight heights of 9 m, 12 m, and 15 m, respectively. LSPIV expresses the LSPIV results from terrestrial fixed stations at a height of 12 m. This indicates that UAV-based LSPIV captures the surface velocity of benchmark measurements using a flow meter at locations 5 and 7 (see Figure 1d), which exhibit higher velocities.

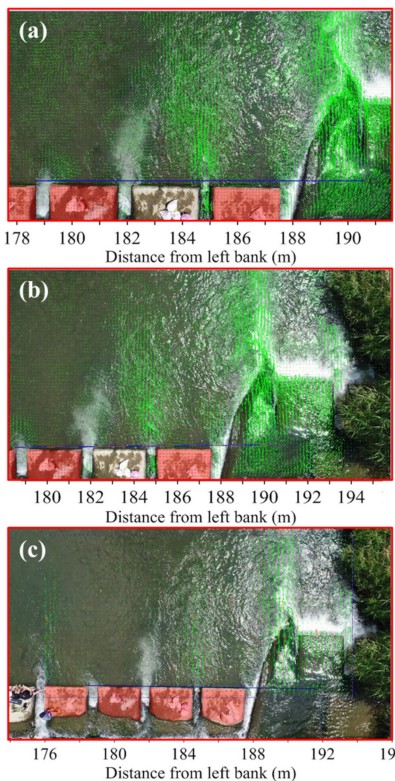

**Figure 5.** Flow field measurements obtained using LSPIV derived from UAVs with different flight heights: (**a**) 9 m, (**b**) 12 m, and (**c**) 15 m. The blue line in figure (**a**) represents the *x*-axis direction.

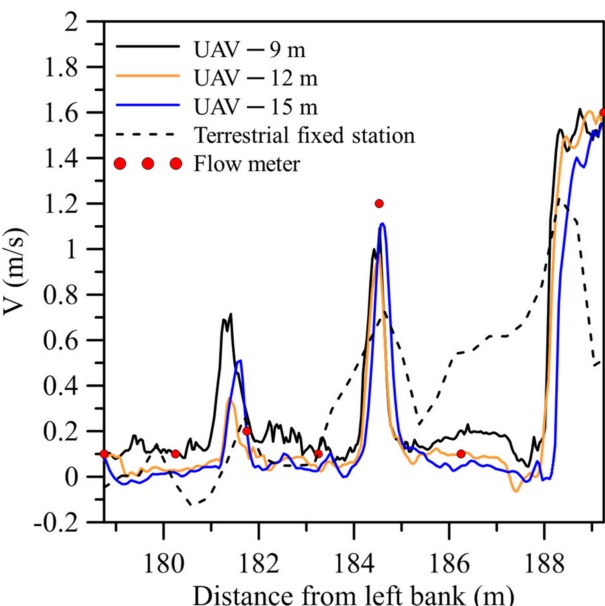

**Figure 6.** Comparison of measured surface velocities using different UAV flight heights, terrestrial fixed stations, and flow meters.

Table 2 presents the resulting statistical errors using different UAV shooting heights and terrestrial fixed stations at the bridge. The measurement results obtained using the flow meter were regarded as the observation data. The mean absolute error (MAE) of the surface velocity measured using LSPIV from terrestrial fixed photography was 0.34 m/s. The MAE values obtained using LSPIV from UAV aerial photography at flight heights of 9 m, 12 m, and 15 m were 0.07 m/s, 0.07 m/s, and 0.04 m/s, respectively. This result indicated that the statistical error of the measured surface velocity using LSPIV derived from terrestrial fixed photography was higher than that using LSPIV from UAV aerial photography [62]. The mean absolute errors of the surface velocity measured by the LSPIV from three different UAV shooting heights were in the range of 0.055 ± 0.015 m/s, exhibiting slight differences from each other.

**Table 2.** The statistical error and the measured surface velocity using LSPIV determined by a UAV held at different flight heights and that of a terrestrial fixed station compared to the results of a flow meter.

| Distance from Left Bank (m) | Flow Meter (m/s) | LSPIV from UAV Flight Height 9 m (m/s) | LSPIV from UAV Flight Height 12 m (m/s) | LSPIV from UAV Flight Height 15 m (m/s) | LSPIV from Terrestrial Fixed Station at 12 m (m/s) |
|---|---|---|---|---|---|
| 178.75 | 0.1 | 0.08 | 0.08 | 0.08 | 0.05 |
| 180.25 | 0.1 | 0.03 | 0.03 | 0.02 | 0.03 |
| 181.75 | 0.2 | 0.12 | 0.12 | 0.19 | 0.05 |
| 183.25 | 0.1 | 0.04 | 0.04 | 0.03 | 0.17 |
| 184.75 | 1.2 | 0.98 | 0.98 | 1.11 | 0.67 |
| 186.25 | 0.1 | 0.10 | 0.10 | 0.10 | 0.55 |
| 189.5 | 1.6 | 1.58 | 1.58 | 1.56 | 0.51 |
| MAE (m/s) | | 0.07 | 0.07 | 0.04 | 0.34 |

The images from the UAV and the terrestrial fixed station have been stabilized/ orthorectified. The terrestrial fixed station produces more errors because the camera is tilted to shoot the images. Although the image can be orthorectified, when correcting the oblique image to an orthoimage, it can be regarded as the compressed distance in the oblique direction and corrected to the correct distance. The extra blank area is filled by interpolation, and there are bound to be errors.

This is why some researchers set up cameras on the bridge to shoot the river surface vertically to yield an orthophoto to avoid orthorectification. In this study, to make the terrestrial fixed station shooting range cover the entire study area, the camera must obtain images by oblique shooting.

When the water flows through the groundsill, complex flow conditions are formed. The water flows through the gaps between the groundsill works, forming fast water flow conditions. The water flow behind the groundsill works is slow. The applicability of UAV-based LSPIV can be further demonstrated by comparing the areas with faster and slower velocities at the same time.

Figure 7 presents the flow fields generated by PIVlab using a UAV flight height of 12 m with different seeding artificial particle densities. The artificial particles (i.e., wood shavings) are highlighted with red square marks during the low flow period. Figure 8 illustrates the measured results of surface velocity using LSPIV from a UAV flight height of 12 m without seeding particles and with different seeding densities. Table 3 indicates the statistical errors with different artificial seeding particle densities and without artificial particles. The MAE values using UAV aerial photography at a flight height of 12 m without seeding particles and with artificial seeding particles of 1.5, 3, and 4.5 kg/min were 0.07, 0.11, 0.11, and 0.12 m/s, respectively. The results indicated that even increasing the density of artificial particles did not significantly enhance the measurement accuracy.

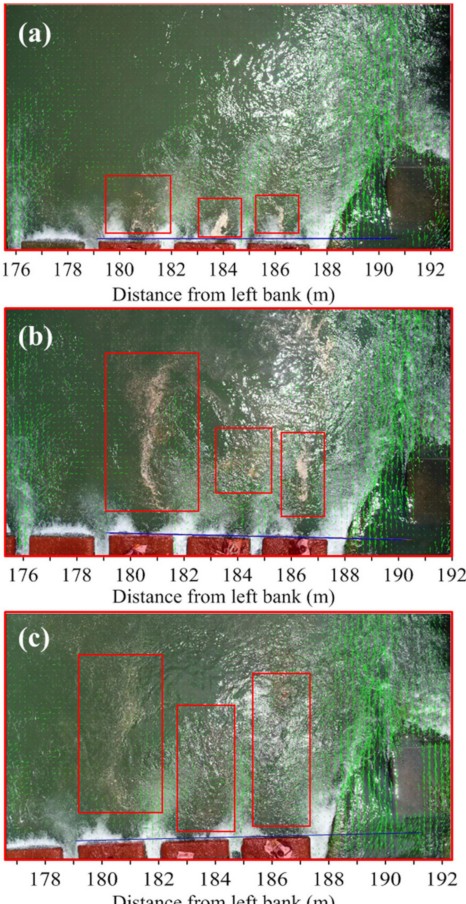

**Figure 7.** Flow field measurements collected using LSPIV derived from a UAV at a flight height of 12 m with artificially seeded particles of (**a**) 1.5 kg/min, (**b**) 3 kg/min, and (**c**) 4.5 kg/min. The blue line in the figure represents the *x*-axis direction. Note that the red square indicates the artificial particles (i.e., wood shavings).

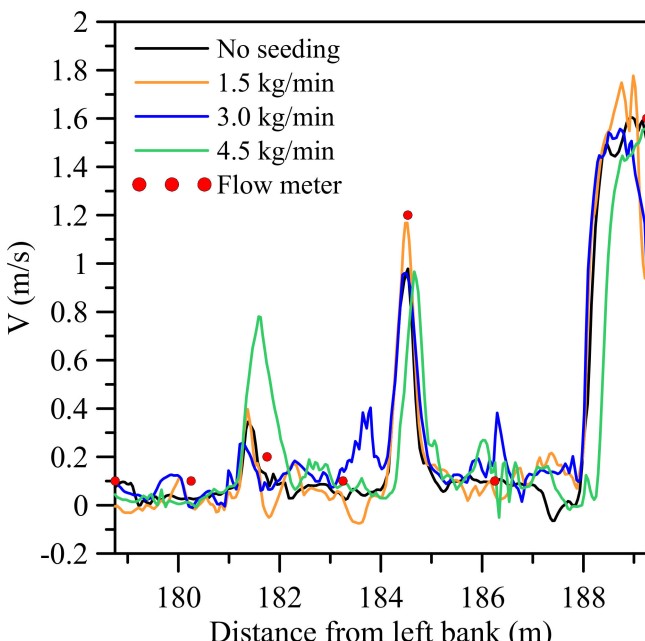

**Figure 8.** Comparison of measured surface velocities obtained using LSPIV derived from UAV flight height of 12 m without seeding particle, with different densities of seeding artificial particles, and a flow meter.

**Table 3.** The statistical error and the measured surface velocity using LSPIV from UAV flight height of 12 m without and with artificial seeding particles.

| Distance from Left Bank (m) | Flow Meter (m/s) | LSPIV from UAV Flight Height 12 m without Seeding Particle (m/s) | LSPIV from UAV Flight Height 12 m with Seeding Particles, 1.5 kg/min (m/s) | LSPIV from UAV Flight Height 12 m with Seeding Particles, 3 kg/min (m/s) | LSPIV from UAV Flight Height 12 m with Seeding Particles, 4.5 kg/min (m/s) |
|---|---|---|---|---|---|
| 178.75 | 0.1 | 0.08 | 0.00 | 0.09 | 0.04 |
| 180.25 | 0.1 | 0.03 | −0.01 | −0.01 | 0.06 |
| 181.75 | 0.2 | 0.12 | −0.04 | 0.08 | 0.59 |
| 183.25 | 0.1 | 0.04 | 0.03 | 0.15 | 0.09 |
| 184.75 | 1.2 | 0.98 | 1.17 | 0.94 | 0.97 |
| 186.25 | 0.1 | 0.10 | 0.05 | 0.25 | 0.18 |
| 189.5 | 1.6 | 1.58 | 1.78 | 1.51 | 1.56 |
| MAE (m/s) | | 0.07 | 0.11 | 0.11 | 0.12 |

### 3.2. Measurement Results on 15 June 2019 under High Flow Conditions

The purpose of the second experiment was to use the UAV's high altitude flight capability as effectively as possible so that the shooting range would cover both banks of the river. To achieve this purpose, the UAV must fly at a height of 112 m. In addition, the camera system with the LSPIV approach set up at the bridge was utilized to measure the surface velocity of the river under high flow conditions. The shooting height of the UAV was gradually reduced so that the UAV shooting width covered half the width of the river and a quarter of the river width. After testing, the UAV met the above requirements at heights of 62 m and 32 m from the take-off location. Due to the action of high flow conditions, a flow meter was not applied for measuring surface flow. Therefore, the float method is an alternative method available to measure surface velocity.

The flow fields generated by PIVlab from UAV flight heights of 32 m, 62 m, and 112 m, and from terrestrial fixed stations at 12 m height are illustrated in Figure 9. This result showed that, due to the high flow conditions occurring on the day of measurement, the

groundsill works were submerged by river water. The measured surface velocity was in the range of 2.065 ± 0.205 m/s. To compare the measurement results using LSPIV at different UAV flight heights and with a terrestrial fixed imaging system, the surface velocity data measured with the float method were regarded as the observation data. Figure 10 compares the measurement results using LSPIV from different UAV flight heights with terrestrial fixed stations and the results using the float method under high flow conditions. In this figure, UAV_32 m, UAV_62 m, and UAV_112 m represent LSPIV analysis completed at UAV flight heights of 32 m, 62 m, and 112 m, respectively. LSPIV denotes the LSPIV results taken by terrestrial fixed stations at a height of 12 m. Table 4 shows the statistical errors of using LSPIV from different UAV shooting heights and terrestrial fixed stations at a height of 12 m. Among them, the UAV aerial shot collected at a height of 62 m using LSPIV for measuring the surface velocity yielded the best result, and the mean absolute error was 0.27 m/s. The mean absolute errors between the terrestrial fixed photography station and UAV flight heights of 32 m and 112 m were 0.46 m/s, 0.49 m/s, and 0.97 m/s, respectively. It should be noted that the UAV flight height of 112 m yielded the worst measurement result. This was the reason that the UAV flight height was high, resulting in a low resolution, making PIVlab misjudge the extent of particle displacement [16,57]. For example, Tauro et al. [57] reported that the image field of view did not exceed 30 × 30 m, and the flight height of the UAV was approximately 30 m.

Based on measurement results of two field campaigns, two new findings with respect to previous research on the topic can be emphasized. The first finding was that the measured results of surface velocity using UAV-based LSPIV were better than those using terrestrial fixed-station-based LSPIV. The second finding was that the tracer can enhance the image recognition using UAV-based LSPIV. However, the tracers were distributed inhomogeneously, leading to limited improvement of the LSPIV measurement results by the tracer.

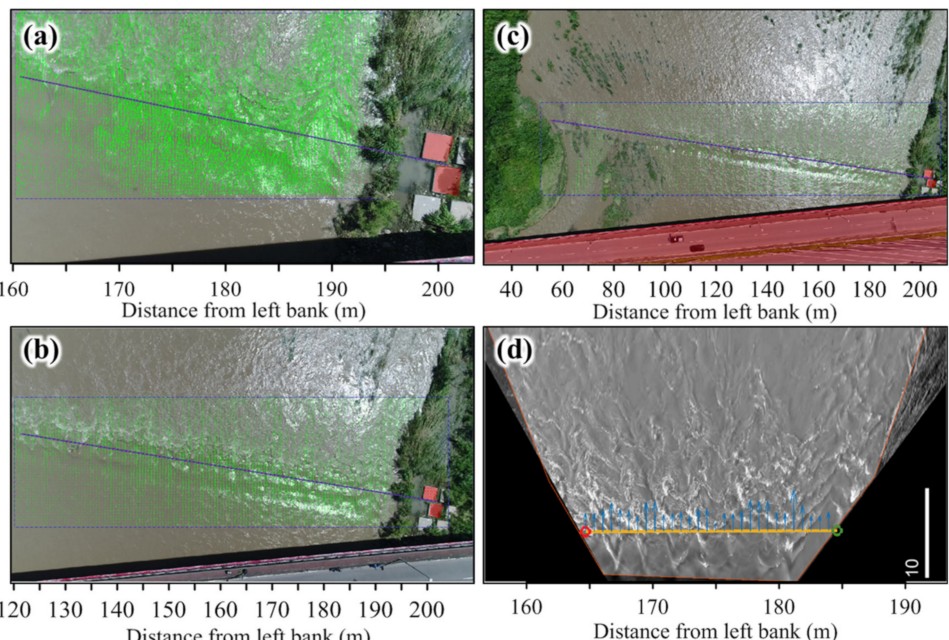

**Figure 9.** Flow field measurements obtained using LSPIV derived from UAVs with different flight heights: (**a**) 32 m, (**b**) 62 m, and (**c**) 112 m. (**d**) The flow field using LSPIV from a terrestrial fixed station at 12 heights. The blue line in the figure represents the *x*-axis direction.

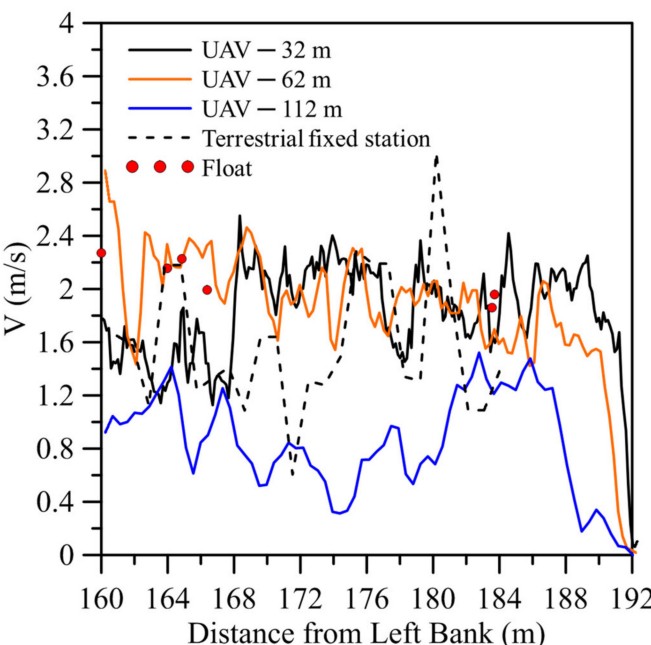

**Figure 10.** Comparison of measured surface velocities using LSPIV from UAV flight heights of 32 m, 62 m, and 112 m, LSPIV from terrestrial fixed stations, and the float method.

**Table 4.** The statistical error and the measured surface velocity obtained using LSPIV derived from a UAV positioned at different flight heights and a terrestrial fixed station compared to that obtained using the float method.

| Distance from Left Bank (m) | Float Method (m/s) | LSPIV from UAV Flight Height 32 m (m/s) | LSPIV from UAV Flight Height 62 m (m/s) | LSPIV from UAV Flight Height 112 m (m/s) | LSPIV from Terrestrial Fixed Station at 12 m (m/s) |
|---|---|---|---|---|---|
| 160.00 | 2.27 | 1.78 | 2.89 | 0.92 | 1.65 |
| 163.98 | 2.16 | 1.29 | 2.33 | 1.36 | 2.18 |
| 164.85 | 2.23 | 1.84 | 2.20 | 1.03 | 2.12 |
| 166.38 | 1.99 | 1.44 | 2.33 | 0.90 | 1.30 |
| 183.55 | 1.86 | 1.61 | 1.64 | 1.24 | 1.24 |
| 183.70 | 1.96 | 1.60 | 1.69 | 1.22 | 1.29 |
| MAE (m/s) | | 0.49 | 0.27 | 0.97 | 0.46 |

## 4. Discussion

Based on the first and second field campaigns, we demonstrated that the UAV-based observations possessed reliable measurement capabilities for measuring surface flow. Moreover, the third field campaign was conducted on 10 December 2019 to explore the influence of the interrogation area and image acquisition time interval on surface flow measurements using LSPIV derived from a UAV. The third field campaign was selected because the hydrological condition represented another lower flow compared to the low flow condition present on 31 May 2019. The hydrological, rainfall, and wind speed conditions observed on 10 December 2019 are described in Section 2.1.

### 4.1. Influence of Interrogation Area (IA) on Surface Flow Measurement

In this field campaign, the UAV was controlled at a flight height of 12 m, with a 1.5 kg/min artificial particle seeding rate. The image acquisition time was set to 1/8 s, and the duration of image shooting was 1 s. Eight frames were used for image analysis. Since the flight height of the UAV is 12 m, the ground resolution is 0.9 cm/pixel. Four interrogation area (IA) cases, i.e., 64 × 64 pixels, 32 × 32 pixels, 16 × 16 pixels, and 8 × 8 pixels, were utilized to measure surface velocity using LSPIV derived from a UAV.

Meanwhile, the measured surface velocity obtained using a flow meter served as the observation dataset.

The IA sizes of 64 × 64 pixels, 32 × 32 pixels, 16 × 16 pixels, and 8 × 8 pixels represented coverage areas of 57.6 × 57.6 cm, 28.8 × 28.8 cm, 14.4 × 14.4 cm, and 7.2 × 7.2 cm, respectively. When tracking artificial particles in more than 10% of the covered IA, they can be considered suitable for use in image recognition to analyze the surface velocity. Approximately 10% of the IA was left empty without artificial particles.

Figure 11 displays the statistical errors of using LSPIV derived from the UAV under different IAs. Because the measurement error of LSPIV with an IA scale of 8 × 8 pixels was greater than that of the IA scale of 16 × 16 pixels, the measurement error did not decrease as the IA scale decreased. The reason was that, when the IA scale was smaller than 16 × 16 pixels, the search range was too small and the number of search features also decreased, which increased the error of image matching, resulting in an increase in the measurement error. The most appropriate IA scale for LSPIV derived from UAVs to analyze the surface velocity was 16 × 16 pixels. Yeh et al. [7] reported that the IA window size should be designed to be greater than four times the particle displacement in PIV, but is more restrictive for LSPIV.

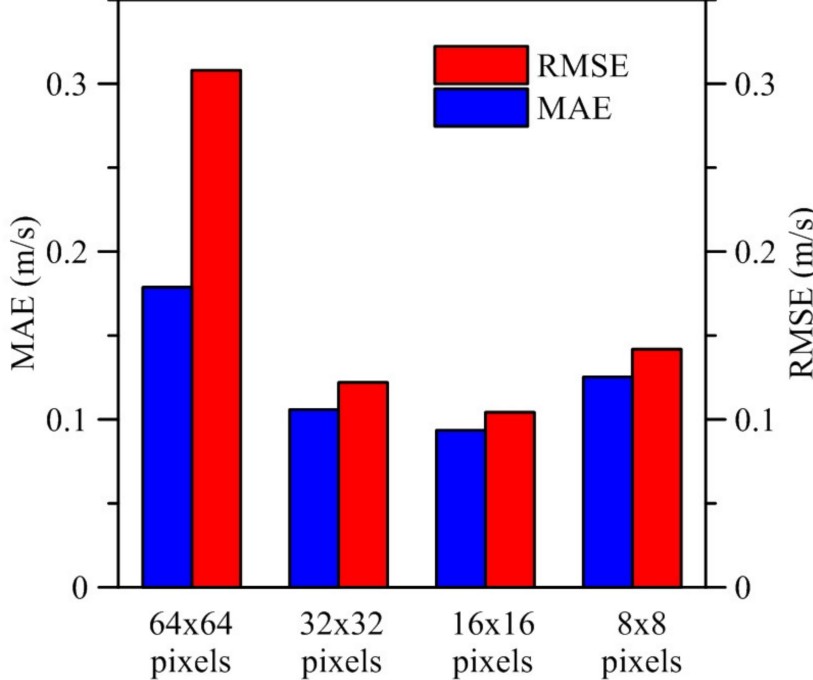

**Figure 11.** The statistical errors of surface velocity for different interrogation areas.

Fujita and Kunita [63] and Yeh et al. [7] used LSPIV with different IA scales to analyze the surface velocity of rivers. They found that small and large IAs would result in large errors in measuring the surface velocity. Fakhri et al. [64] documented that the optimum IA value in contacted open channel flows was 60 × 60 pixels. The IA scale selected in LSPIV from UAV is adapted to local conditions, so the IA scale should be selected carefully to reduce the error associated with measuring the surface velocity.

### 4.2. Influence of the Image Acquisition Time Interval on the Surface Flow Measurement

The image acquisition time interval of the recorded images has a direct influence on the accuracy of the surface velocity measured using LSPIV from UAVs. The flow condition on 10 December 2019 represented a low flow period. To demonstrate that low flow conditions require a lower frame rate than high flow conditions, four different image acquisition time intervals, including 1/8, 1/4, 1/2, and 1 s, corresponding to 96, 48, 24, and

12 frames, respectively, were adopted to explore the measured surface velocity results. The IA window size was set to 16 × 16 pixels.

Figure 12 depicts the statistical errors of using LSPIV derived from UAVs under different image acquisition time intervals. When the image acquisition time interval increased, the statistical errors of the MAE and RMSE values increased. This was attributed to the fact that the tracers might leave an interrogation area between images. This resulted in high statistical errors in the measured surface velocity. Similar results were also demonstrated by Meselhe et al. [65] and Dobson et al. [4].

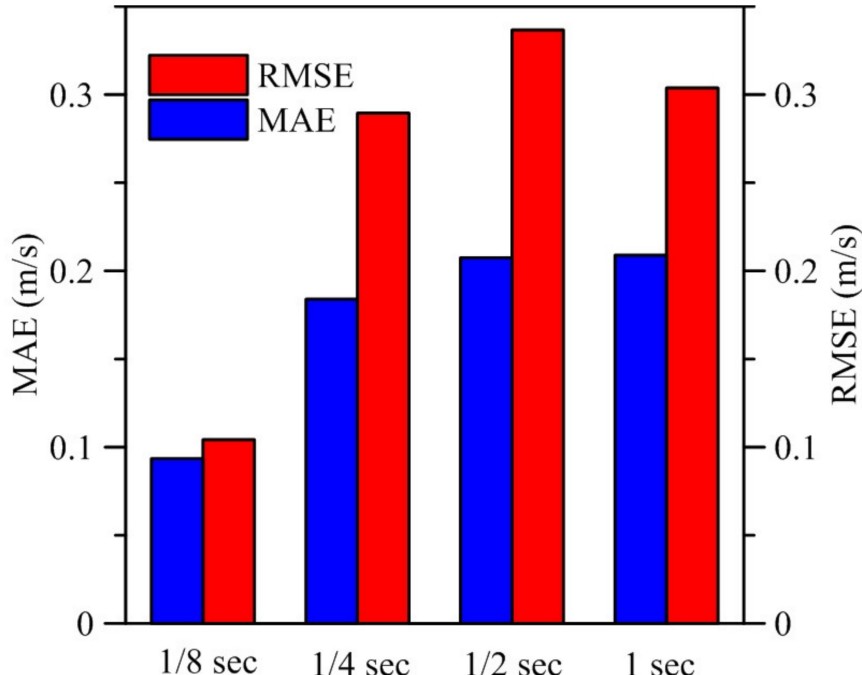

**Figure 12.** The statistical errors of surface velocity for different image acquisition time intervals.

Although the image acquisition time interval increases, the images can be distinguished and theoretically analyzed during the low flow period. However, if the image acquisition time interval is too long, resulting in the lack of waves on the river surface and consecutive tracer patterns, this would lead to less accuracy in surface velocity analysis using LSPIV.

### 4.3. Relevant Discussion, Limitations, and Future Work

This study explores the application of UAV-based LSPIV in Taiwan, since it is a reliable nonintrusion measurement technique. Although the research possesses a site-specific case study, it demonstrates the possibility to utilize the presented measurement technique under high flow conditions, especially during flooding.

UAV flight heights at 9 m, 12 m, and 15 m were employed to analyze the surface velocity, corresponding to pixel mapping to ground distances of 0.7, 0.9, and 1.2 cm/pixel, respectively (see Table 1). The MAEs were 0.07, 0.07, and 0.04 m/s (Table 2). Lewis et al. [54] pointed out that, when the flight height ranged from 9.4 to 35 m, the pixel mapping to ground distance was between 0.35 and 1.45 cm/pixel, and the measured average velocity was between 0.09 and 0.14 m/s. The lowest average velocity measured at a flight height of 27.4 m was 0.09 m/s, which indicated that the surface velocities analyzed at different heights were not very different from each other. Therefore, the statistical error (MAE) between the surface velocities measured by UAV-based LSPIV and the benchmark measurement did not increase with increasing flight heights.

The function of the tracer is to enhance the pattern of the river surface in the image, so this does not mean that the difference between the LSPIV analysis and the benchmark measurement surface velocities will be significantly reduced after seeding the tracer, especially during low flow periods. When the UAV flight height is 9 m, the ground distance represented by 1 pixel is 0.7 cm. In this study, the IA is set to $16 \times 16$ pixels. When the UAV flight height is 9 m, the IA coverage area is $10.2 \times 10.2$ cm (see Table 1). Since the size of the wood shavings (i.e., artificial tracers) is between 0.25 and 6 cm$^2$, the selected IA size can distinguish the difference between wood shavings and no wood shavings in rivers. When the UAV flight height rises to 112 m, the distance from the ground represented by 1 pixel also increases to 9 cm. The size of 1 pixel is larger than the size of the wood shavings; therefore, the wood shavings may not be truly projected in the image, leading to the error in analyzing surface velocity increasing significantly. When the size of a pixel can no longer display wood shavings, regardless of the IA coverage area, it is impossible to distinguish wood shaving areas.

Experiments of artificially seeded tracers were utilized to simplify the identification of moving patterns on the water surface, especially under low flow conditions [66]. Pizarro et al. [56] used particle tracking velocimetry (PTV) and particle image velocimetry (PIV) to measure stream surface flow. A tracer seeding metric called the seeding distribution index (SDI), based on seeding density and trace clustering, was introduced to identify the optimal spatial distribution of tracers. They found that the error reductions were 15.9% and 16.1% using PTV and PIV, respectively. Pizarro et al. [67] also introduced the novel idea of optimizing image velocimetry by selecting two reasonable criteria, i.e., the maximization of the number of frames and the minimization of the dimensionless SDI. The shooting frames were analyzed using LSPIV to estimate the surface flow velocities and river discharge rates. They concluded that the proposed framework might reduce the errors of the computed discharge errors by 0.4 and 0.12% for two study cases in rivers. Meselhe et al. [65] suggested that a seeding particle density of 10–30% of the surface flow area was a good guideline to adopt in various field applications using LSPIV. In this study, the experiments with wood shavings revealed that the selected tracer size was too small and that the tracers were distributed inhomogeneously. Therefore, tracer seeding did not improve the LSPIV results.

The specific error associated with rectification can be controlled by ensuring that the camera lens is orthogonal to the water surface and that the results are corrected for distortion [9,53,62]. However, it is not always possible to keep the camera lens orthogonal to the water surface when the UAV moves to capture the surface flow. This means that camera motion can diminish the measured precision of surface velocity [10]. Perks et al. [11] adopted fixed control points that were manually selected and automatically tracked between frames. Lewis and Rhoads [15] stated that UAVs were stable enough to achieve unbiased mean velocity measurements within a 30-s sample under an average wind speed of up to 5.2 m/s under low flow conditions. Because ADCP is not suitable for shallow water bodies or obstacles in rivers, the flow meter and float method are utilized for benchmark measurements. However, these measurement methods contain uncertainty in their accuracy. These uncertainties in the surface velocity measurement are still worthy of further investigation.

Regarding the issue of image stabilization, UAV operation at great heights leads to an increase in the magnitude of camera movement along the ground distance [15], such that even subpixel apparent motion may introduce errors into the flow velocity calculation. Tauro et al. [57] mentioned that, as the number of analyzed images increases, the degree of interference of image shaking gradually decreases. Because the direction of image shaking is irregular, when there are more images, the flow velocities generated by different shaking directions between different images eliminate each other as a result of increasing the analyzed images. Tauro et al. [58] also investigated the influence of UAV vibration on surface velocity measurements. They found that the use of a three-axis gimbal with the camera mounted on the UAV effectively reduced the vibration of the UAV. They also demonstrated that UAVs could be utilized to obtain accurate surface flow

maps of submeter water bodies. Therefore, UAV vibration does not hinder surface velocity observations in rivers.

The literature indicated that PIV did not work well under unseeded conditions [53,55,56, 67]. Other techniques, such as space–time image velocimetry (STIV) [68] and surface structure image velocimetry (SSIV) [69], are much more suitable if there are no materials or artificial tracers in the flow. In the present study, all of the images display surface velocity fields with a great number of spurious vectors in areas where visible structures on the water surface are absent. If surface velocity measurements utilized for validation were performed in these areas, worse comparison results would be yielded.

With current technology, it should not be problematic to measure surface velocity with UAVs, but UAVs need to be further enhanced because they cannot fly under strong wind and heavy rainfall conditions, including those of typhoons, nor do they have a continuous power supply, which is another issue that needs to be resolved. How to transfer the images taken by UAVs to the computer and directly calculate the flow field through LSPIV to achieve the goal of automation is another challenge.

There are several potential image velocimetry algorithms, such as Kanade–Lucas–Tomasi image velocimetry (KLT-IV), large-scale particle tracking velocimetry (LSPTV), optical tracking velocimetry (OTV), STIV, and SSIV, that can be utilized to analyze the surface flow [11,39,45,69]. In future research, more UAV algorithms will be developed to explore their pros and cons and determine which algorithm is suitable for local conditions to measure the surface velocity of rivers.

## 5. Conclusions

To demonstrate the capability of UAVs for measuring flow fields in rivers, the measured surface velocities in the Houlong River of central Taiwan using LSPIV from UAVs and from terrestrial fixed stations were compared. Two field campaigns were conducted on 31 May and 15 June 2019, representing low flow and high flow conditions, respectively, to measure the surface velocity. The IA window size and image acquisition time were set to $16 \times 16$ pixels and $1/8$ s, respectively. The duration of image shooting was 1 s. This means that 8 frames collected within 1 s of image shooting were used for analysis. The images from the UAV and the terrestrial fixed station were stabilized/orthorectified for further analysis. The results indicated that the mean absolute errors of the surface velocity measured by LSPIV from three different UAV shooting heights (i.e., 9 m, 12 m, and 15 m) ranged from $0.055 \pm 0.015$ m/s during low flow conditions, which indicated slight differences from each other. However, the measured results using LSPIV derived from UAVs were better than those using LSPIV derived from terrestrial fixed stations on bridges. The measured surface velocities obtained using LSPIV from UAVs with different densities of artificial seeding particles (i.e., wood shavings) were compared to those collected without artificial seeding particles, demonstrating that increasing the density of artificial particles insignificantly enhanced the measurement accuracy. This would be the reason that the tracer size was too small and the tracers were distributed inhomogeneously, resulting in the insignificant improvement of the LSPIV results by the tracer.

The measured surface velocities using LSPIV from UAVs with different shooting heights (i.e., 32 m, 62 m, and 112 m) were compared under high flow conditions. This result indicated that a UAV flight height of 112 m yielded the worst measurement result because of the issue of low image resolution.

The third field campaign was conducted on 10 December 2019, during the low flow period to investigate the impact of the interrogation area (IA) and image acquisition time interval on surface flow measurements obtained using LSPIV performed by a UAV. The results indicated that the most appropriate IA scale and image acquisition time interval using LSPIV derived from UAV to analyze the surface velocity were $16 \times 16$ pixels and $1/8$ s, respectively. These two parameters should be selected carefully according to local conditions, because they significantly affect the surface velocity measured using LSPIV derived from UAVs.

Finally, we would like to remind that the main contribution of this study is the value of practical application, not entirely emphasized on novelty.

**Author Contributions:** Conceptualization, W.-C.L.; methodology, C.-H.L. and W.-C.H.; software, C.-H.L.; validation, W.-C.L., C.-H.L., and W.-C.H.; formal analysis, C.-H.L.; investigation, W.-C.L.; resources, W.-C.L.; data curation, C.-H.L. and W.-C.H.; writing—original draft preparation, W.-C.L.; writing—review and editing, W.-C.L. and W.-C.H.; visualization, C.-H.L.; supervision, W.-C.L.; project administration, W.-C.L.; funding acquisition, W.-C.L. All authors have read and agreed to the published version of the manuscript.

**Funding:** This study was partially supported by the Ministry of Science and Technology (MOST), Taiwan, under grant no. 109-2625-M-239-002.

**Institutional Review Board Statement:** Not applicable.

**Informed Consent Statement:** Not applicable.

**Data Availability Statement:** Data and images are available at website, https://github.com/e11856 824/LSPIV-with-UAV/ (accessed on 4 June 2021).

**Acknowledgments:** The authors would like to express their appreciation to the Ministry of Science and Technology (MOST) for providing the funding support. The authors sincerely appreciate three anonymous reviewers who provided useful comments to improve this article.

**Conflicts of Interest:** The authors declare no conflict of interest.

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
