# Peer review of "Large-Scale Particle Image Velocimetry to Measure Streamflow from Videos Recorded from Unmanned Aerial Vehicle and Fixed Imaging System"

_remotesensing, doi:10.3390/rs13142661_

Round 1
Reviewer 1 Report
Dear Authors,
many details are included and the paper have certainly improved from the first version. However, I have still some doubts about it, since my main comment is not properly assessed: it is not clear to what extent the paper is presenting some new findings respect to previous research on the topic.
The manuscript is still dominated by the presentation of a case study and, it is difficult to evaluate the scientific soundness of the work and to generalize the results. I agree that this contribution should have practical value for this topic but can be not considered totally a novelty.
Other comments:
Respect to the answers provided and the descriptions included in the main text, please focus on the settings used for your application rather than on generalized and didactic contents (e.g. answer to points 3); in other cases, please move general background parts (e.g. answer to point 10) in the introduction section or in the discussion if it can be useful for comparative analysis.
Figure 4: I think that this figure is not strictly necessary.
Section 3: This section is still dominated by the description of the three hydrological events and of the methods and material used rather than to the results.
Author Response
Please see the atteched file.

Reviewer 2 Report
GENERAL COMMENTS
This manuscript is a resubmission of the manuscript with ID:1215920. The latest version appears significantly improved. However, some issues remain to be addressed.
- The Title needs to become more compact. A suggestion is "Large-Scale Particle Image Velocimetry to measure streamflow from videos Recorded from Unmanned Aerial Vehicle and Fixed Imaging System"
- Some references need to be added (see specific comments).
- There are some typos (see specific comments).
- A web page has been created, but the dataset has not been uploaded to it to become publicly available. Please upload the dataset (video or frames) and a metadata file.
SPECIFIC COMMENTS
Location: "... large-sale particle image velocimetry ..."
Comment: It should be "large-scale", not "sale".
Location: "... yielding much better agreement with the results of field measurements."
Comment: Add text and reference "... yielding much better agreement with the results of field measurements. Various researchers [49, 50] have suggested employing Monte Carlo simulations to both estimate the uncertainty of the obtained LSPIV estimations and eliminate the subjectivity on the selection of the LSPIV algorithm parameters (e.g. interrogation area size, size of the contrast-limited adaptive histogram equalization, etc.)."
REFERENCES
49. Le Coz, J., Renard, B., Vansuyt, V., Jodeau, M., & Hauet, A. (2021). Estimating the uncertainty of video‐based flow velocity and discharge measurements due to the conversion of field to image coordinates. Hydrological Processes, 35(5), e14169.
50. Rozos, E., Dimitriadis, P., Mazi, K., Lykoudis, S., & Koussis, A. (2020). On the Uncertainty of the Image Velocimetry Method Parameters. Hydrology, 7(3), 65.
Author Response
Please see the attached field.

Round 2
Reviewer 2 Report
The title has an error: "to Measurement" -> "to Measure"
Author Response
The title has an error: "to Measurement" -> "to Measure".
Response: The title has been revised as “Large-Scale Particle Image Velocimetry to Measure Streamflow from Videos Recorded from Unmanned Aerial Vehicle and Fixed Imaging System”.
This manuscript is a resubmission of an earlier submission. The following is a list of the peer review reports and author responses from that submission.
Round 1
Reviewer 1 Report
General comment: This paper focus on the application of LSPIV technique to monitor surface flow velocities from UAS platform and fixed station in rivers. The topic is interesting for readers and it can help technicians to identify adequate measurement settings. However, at the moment, the presentation of the methodology and experimental design need more details. Only in this way it is possible generalize the outcomes of this research. My opinion is that the paper needs a big improvement before being reconsidered for discussion and eventually, publication. In the following, I list some specific comments that should be addressed carefully.
Specific comments:
1) Section 1: Introductory section seems definitely too generic, and does not focus on the core of the topic that is the rule of shooting height, artificial seeding tracer, interrogation area (IA), and acquisition time on LSPIV accuracy. Several other references could be used to give a more complete framework of the topic and to emphasize the novelty of this work.
2) Section 2.2: Seems that the authors use a fixed camera with an horizontal angle respect to the ROI selected. Why do you use this configuration? Please describe and motivate this choice.
3) Section 2.4: No information is provided on pre and post-processed steps of the analysis. If the images are pre-processed, the inclusion of an example of a pre-processed image can be useful to understand the appearance of the tracers and the wave patterns on the free surface.
Section 2.5:
4) The description of benchmark measurements require more details, especially respect to the float method.
5) The measurement locations used for the velocity comparison should be included in the figure.
Section 3:
6) The description of the three hydrological events and all the information on the three field campaigns should be included in the previous sections “Materials and Method”.
7) It is interesting the attempt to refer the accuracy velocity reconstruction using LSPIV to local seeding condition. Unfortunately, the information on seeding is provided in terms of bag/min. Please include a more quantitative information.
8) Authors should specify the shape and dimension of tracers used in their experiments. The size of tracers is an important criterion that influences the recognisability of individual traceable features in the acquired optical data depending on their spatial resolution.
9) It is note that UAS operation at high heights also leads to the increase of the magnitude of camera movement in ground distance (Lewis & Rhoads, 2018), such that even sub-pixel apparent motion may introduce errors into flow velocity calculation. Moreover increasing the flight height, the effect of wind can represent an issue. How is it possible avoid this issue? Do you have performed image stabilization? Please include this part in the discussion section.
10) Authors write: “The results indicated that even increasing the density of artificial particles did not significantly enhance the measurement accuracy [46].” Please motivate these findings in the discussion section. Most of the literature experiments were artificially seeded to simplify the identification of moving patterns on the water surface, especially during low flow conditions (Perks et al., 2020).
Section 4:
11) Please motivate the use of the third field campaign for investigating the role of time interval and interrogation size on LSPIV accuracy.
12) The discussion section should include the other settings considered in this study: shooting height, artificial seeding tracer.
Reviewer 2 Report
Dear authors,
I have carefully studied the results of your research. It is great that you are exploring the application of the UAV based LSPIV in Taiwan since it is a reliable non-intrusive measurement technique. At the same time, the quality of LSPIV results strongly depends on the correctness of its application. I see major issues in the way you applied LSPIV and validated the analysis results. They must be improved before publishing your research.
- Lines 159-161: the principle of PIV is not described correctly. PIV is used to analyze the displacement of particle groups – not individual particles – based on cross-correlation algorithms. Initially, PIV determines flow velocity (magnitude and direction) in pixels per frame. I can recommend you the following publications which describe the principles of PIV:
- Adrian, R.J. Twenty years of particle image velocimetry. Experiments in Fluids 2005, 39, 159–169, doi:10.1007/s00348-005-0991-7.
- Raffel, M.; Willert, C.E.; Scarano, F.; Kähler, C.J.; Wereley, S.T.; Kompenhans, J. Particle image velocimetry: A practical guide, Third edition; Springer: Cham, 2018, ISBN 978-3-319-68851-0.
- Line 358: you state that “LSPIV adopts a traditional cross-correlation algorithm”. This statement shows that your understanding of PIV principles needs to be improved. PIVlab alone employs 3 different cross-correlation algorithms for LSPIV: DCC, FFT and ensemble cross-correlation. Which of them is "traditional"?
- You have selected the image resolution of 1920 x 1080 px in order to make the UAV based footage comparable to the terrestrial footage. This image resolution may be considered suitable for low heights (9 – 12 m). However, it is absolutely not applicable to images acquired at 112 m height. The fact that LSPIV analysis of the footage captured at 112 m yielded the worst results is not a scientific conclusion. You must inform the reader about the ground sampling distance in each of the acquired image sequences/videos. You must describe the size of visible structures on the water surface in real-world units and in pixels to justify the selected IA size. If you use the same PIV settings for different ground sampling distances, it is a proven fact that the quality of analysis will deteriorate with growing flight height, and it does not need to be proven again.
- It is a proven fact that PIV does not work well in unseeded conditions. Other techniques, such as STIV and SSIV, are much more suitable if there are no natural or artificial tracers in the flow. All of your images illustrate surface velocity fields with great numbers of spurious vectors in the areas where visible structures on the water surface are absent. If flow measurements used for validation were performed in these areas, you would get much worse comparison results. This issue must be addressed in the text of the paper.
- Images which illustrate experiments with wood shavings show that tracer size is too small and that tracers are distributed inhomogeneously, therefore such seeding even in principle cannot improve the LSPIV results. Your conclusion that “increasing the density of artificial particles did not significantly enhance the measurement accuracy” is wrong both in terms of the description of your study results (the highest seeding density is associated with the greatest error) and in terms of improper methodology. Please consult the following papers on seeding density and distribution:
- Pizarro, A.; Dal Sasso, S.F.; Perks, M.T.; Manfreda, S. Identifying the optimal spatial distribution of tracers for optical sensing of stream surface flow. Hydrol. Earth Syst. Sci. 2020, 24, 5173–5185, doi:10.5194/hess-24-5173-2020.
- Pizarro, A.; Dal Sasso, S.F.; Manfreda, S. Refining image‐velocimetry performances for streamflow monitoring: Seeding metrics to errors minimization. Hydrological Processes 2020, 34, 5167–5175, doi:10.1002/hyp.13919.
- You must add the following information to the text:
- How many images have you used for each of the measurements?
- Did you subsample or use the original frame rate?
- How did you aggregate the results?
- How did you extract the results from the PIVlab for comparison? How did you make sure that measurement locations of validation measurements correspond to the correct location in the PIVlab velocity field?
- Tables 1 and 2 show the same reference measurements performed with the flow meter. However, Figures 4 and 6, which must include the same reference measurements, illustrate great differences between the flow meter measurements. In the Figure 4, four of the flow meter measurements are less than 0.1 (visually - almost 0), and none of the measurements is 1.2 or 1.6 m/s which are referenced in the corresponding Table 1. Either Tables or Figures are wrong.
- Did you stabilize your footage before processing? Can your data be seen in order to visually assess whether it requires stabilization?
- The validation dataset is not convincing. Why have you selected reference measurements, most of which are recorded behind obstacles and therefore close to 0, and performed them with a device with low accuracy, which makes recording of subtle differences in velocity impossible? 4 out of 7 available reference measurements equal 0.1 m/s, which, given the flow meter accuracy, can be anything from 0 to 0.2 m/s. How do you account for these uncertainties when analyzing the differences between the LSPIV results and the reference measurements?
- Line 321: without informing the reader about the ground sampling distance and the size of tracers, IA size itself is useless. How many particles fit on average into each IA in your comparison? How many IAs (%) are left empty, without particles?
- Line 331: “caused too many vectors to be searched” – this is a wrong statement.
- Which cross-correlation algorithm have you employed? Single- or multi-pass?
- Line 346: suitable frame rate depends on flow conditions; low flow requires lower frame rate than high flow. Which flow have you measured? How did you select 4 frame rates for comparison?
- Conclusions must be rewritten considering all the issues specified.
Less significant issues:
- Lines 19-20: what do you mean by “using UAV with a terrestrial fixed station”?
- Line 58: “orthorectify imagery with known GCP coordinates” – orthorectification can also be performed based on known distances between the GCPs if coordinates are unknown or cannot be measured accurately.
- Line 60: “optical flow technologies” is a wrong term as “optical flow” is a method which you neither study nor use in your research. Use “optical methods of flow analysis” or “optical methods of flow measurement” instead.
- Line 69: rephrase “LSPIV also provides a flexible and cost-effective flow monitoring method”. LSPIV is a method itself, it does not provide a method.
- Line 76: reference [5] is not Tauro et al.
- Lines 77-78: what do you mean by “UAV photography can overcome the barriers of orthophotos”? UAV footage is often converted to orthophotos before measurements because it has to be distortion-free and characterized by uniform scale in order to enable accurate measurements.
- Line 79: pioneering research in combining PIV and UAVs was conducted in the same 2015 by M. Detert, who further published the 2017 study which you cite as [13].
- Detert, M.; Weitbrecht, V. A low-cost airborne velocimetry system: proof of concept. Journal of Hydraulic Research 2015, 53, 532–539, doi:10.1080/00221686.2015.1054322.
In 2021 he has published another study worth reading in order to avoid errors when applying LSPIV
- Detert, M. How to Avoid and Correct Biased Riverine Surface Image Velocimetry. Water Resour. Res. 2021, 57, 23, doi:10.1029/2020WR027833.
- Groundsill is written inconsistently (both together and separately)
- Study site is not sufficiently described (explicitly state the river width + the size of the field of view for each image sequence)
- Line 165: what do you mean by “need to convert to continuous frames when shooting images”? Converting a video into an image sequence? Subsampling?
- Line 166: Statement makes a false impression that RIVeR is a PIVlab toolbox. It is not. It is a separate software and works with the PTVlab results to the same extent as it can work with PIVlab results.
- Line 167-168: orthorectification is not used to reduce the computational effort, it is a must for an accurate measurement from oblique images and cannot be avoided independent on the computational cost.
- For the float method: which floats were used, over which distance and time, how many times were the measurements repeated?
- For the flow meter: how long was each measurement taken? Was it repeated and averaged?
- Line 218: Why do you use the verb “poured”? Were wood shavings soaked in water before introducing them into the flow? If so, it must be stated in the text.
- Line 219: how much is one bag in kg and l/m3?
- Lines 237-238: “statistical error … from terrestrial fixed photography was higher than that using LSPIV from UAV aerial photography”: how do you explain this fact? If no stabilization/orthorectification was performed for the UAV footage, and terrestrial footage was orthorectified correctly, it is a very strange result. Can it be explained by the differences in the field of view? Did you assess possible orthorectification errors by terrestrial imagery?
- Line 243: what do you mean by “mainly highlighted in the flow field with low flow areas”?
- Figure 5: blue line is present in (a) and (b) and absent in (c). Why?
- It will be useful to introduce the x axis values (in m) into the Figures 3 and 5, because the vector field representation without color-coding and image quality do not facilitate easy comparison of locations between Figures 3-4 and 5-6.
- Lines 367-368: Lewis and Rhoads did not claim that no stabilization is needed “under the windiest condition”, they talked about the average wind speed of up to 5.2 m/s. Even this statement, however, has a limited application, especially in low-flow conditions.
Reviewer 3 Report
GENERAL COMMENTS
This manuscript regards a study on the accuracy of the LSPIV method applied to videos recorded from a fixed position and videos recorded from UAVs. There are some serious issues that need to be addressed:
- The manuscript language suffers from incorrect word usage (see specific comments).
- The manuscript does not follow standard scientific format. For example, there are references to other studies in the last paragraph of Conclusions.
- Details are missing regarding some of the employed methodologies (e.g., float method).
- The study appears as a report of a site-specific case study without broad scientific interest.
- Some references to relevant studies are missing.
For these reasons, I cannot recommend the publication of this manuscript. I would suggest to the authors to resubmit after they do the following: first, have the text edited by a native English speaker, second, and most importantly, make the data (videos and flow meter measurements) publicly available to increase the scientific value of the manuscript.
SPECIFIC COMMENTS
Location: Title
Comment: Slightly change the title to "Surface Flow Measurements Using Large-Scale Particle Image Velocimetry to Videos recorded from Unmanned Aerial Vehicle and Fixed Imaging System"
Location: "River velocity and flow play an important role in the effective management of water resources, which relies on accurate measurements."
Comment: Change to "The accuracy of the measurements of the river velocities plays an important role in the effective management of water resources."
Location: "... the results obtained using UAV with a terrestrial fixed station"
Comment: The meaning is not clear. Do you mean "using LSPIV on videos recorded from a terrestrial fixed station"?
Location: "The float method divides the distance between two sections by the time difference between the floats passing through the two sections to obtain the average velocity."
Comment: Is the float method a standard method? If so, a reference is required. If it is an improvisation, more details are required to describe it.
Location: Figure 4
Comment: Surprisingly, there are 5 measurements at positions with stream flow velocity lower than 0.1 m/s, which is below the operating velocity range (see line 179) of the flow meter. On the other hand, no measurement was taken at the location of 188 m from the left bank, where the flow is significant.
Location: Figure 6
Comment: The 'UAV_12 m' in the legend should be replaced with 'No seeding'.
Location: Figure 8
Comment: The 'LSPIV' in the legend should be replaced with 'Terrestrial fixed station'.
Location: "To demonstrate the availability and capability of UAVs for measuring flow fields ...
Comment: Whether UAVs are available or not (in the market or affordable to buy) is an issue that does not concern the scientific community.